# The Assessment of Innovation Development in the Arctic Regions of Russia Based on the Triple Helix Model

**Nikolay Egorov [1]** , **Tatiana Pospelova [2]**, **Anastasia Yarygina [3]** and **Elena Klochkova [4],***

[1] Scientific-Research Institute of Regional Economy of the North, North-Eastern Federal University, Yakutsk 677000, Russia; ene01@yandex.ru

[2] Economic Department, Lomonosov Moscow State University, Moscow 119192, Russia; pospelova_t@mail.ru

[3] Innovation Department, Hyundai Motor Company, Seoul 137938, Korea; yar.anastasia@gmail.com

[4] Center for Monitoring Science and Education, Peter the Great St.Petersburg Polytechnic University, St. Petersburg 195251, Russia

* Correspondence: samarinne@list.ru

**Abstract:** This article presents the methodology and tools for the econometric analysis and assessment of the innovation development of the Russian Arctic regions, under the Triple Helix concept. The econometric calculations based on this methodology allow the express assessment of innovation development of a region and the contribution of the science and education complex (SEdC), industries, and the government, to the integral index of innovation development of this region, using minimum key statistical indicators in the field of science and innovation. The calculation results obtained using the author's methodology show the adequacy of the express rating, with respect to the rating data obtained using other methodologies. The main advantage of the proposed methodology is the elimination of the human factors arising from the weighing coefficients and the results of the expert assessments used in the other rating methodologies. The calculation results obtained using this methodology might be useful to the executive bodies of state authorities, business entities, and scientific and educational institutions, for an express assessment and for making various organizational and managerial decisions on innovation development in a region.

**Keywords:** Triple Helix; methodology; econometric model; express assessment; innovative products

## 1. Introduction

The efficiency of innovative policy implementation depends largely on the system of indicators that provide the grounds for the detection of innovative activity, and the monitoring of its development. In this regard, one of the main tasks in this direction is to form a set of indicators to assess the level of innovation development of a region, which would take into account the necessary capabilities and resources [1–4]. However, there are no universal approaches to assessing the level of region innovation development in managerial practice, consequently, this impedes an adequate assessment of the effectiveness of state innovation policy, at the federal and regional levels, and the effectiveness of budget spending [4]. Russian researchers have dissimilar opinions on this issue. In general, the analysis of relevant literature shows that the main reason for the existence of such a variety of methodologies is the lack of a unified methodology for choosing indicators that characterize innovative potential [5–9]. Assessments of regional innovative potential are mainly conducted on the basis of expert survey data, which introduces subjectivity of indicators that leads to inaccuracy of the assessment results. In this regard, we present an author's model and methodology to perform econometric assessments

of the level of an economic entity innovation development (EEID), based on the Triple Helix model concept [10–13].

## 2. The Econometric Model to Assess the Level of EEID

The success in building a modern progressive society, based on a knowledge of economy, is impossible without reasonable management of innovation development and its modeling. For this reason, the need to elaborate innovation development models has nowadays grown tremendously, including the "model" in its narrow sense, i.e., a physical and mathematical or other analogue of the innovation development process. In the Triple Helix model, each helix represents an independent process and has unique properties, as well as its own metrical parameters derived from specific values [14–17]. Thus, the principles of bibliometrics and scientometrics with relevant measurement indicators such as the number of publications in peer-reviewed journals and citation indices, the number of applications for patents, and the number of patents received are applicable for the *U*-component (university). The *B*-component (business) is usually understood as technology business (industry) focused on the introduction of high-tech products coming from the *U*-component into the industry. The analysis of the state political activity and its impact on the development of the *U*-and *B*-components of the Triple Helix is important to study the *G*-component (government) [18].

Currently, there are various methods and models for assessing the level of regional innovation development in Russia [19]. At the same time, there is no data on any quantitative methods to assess the contribution of the science and education complex (SEdC), industry, and government, to the integral innovation development of an economic entity in the foreign and domestic literature on economy. In this regard, author N.E. Egorov elaborated a method of integral assessment of the EEID level, based on the well-known Triple Helix model [10–13]. The formation of an effective innovation system is possible upon the achievement of simultaneous paired harmonic relations, of science with business, government with science, and government with business that builds up a special environment, a dimensional space of innovations promoting the creation and spread of innovations [20–22]. Ideally, it takes the shape of a cubic volume, but in real conditions of regional economic development this innovation space can take various shapes of a rectangular parallelepiped, depending on the degree of relations between the science and education complex (SEdC), industry, and government (Figure 1). The econometric Triple Helix model presented, allows the quantification of the contribution of each of the triad participants, to the innovation development of economic entities of different levels, based on known trigonometric expressions. Let us note that a similar model in the form of a vector representation of university–industry–government relations is discussed in articles [23,24].

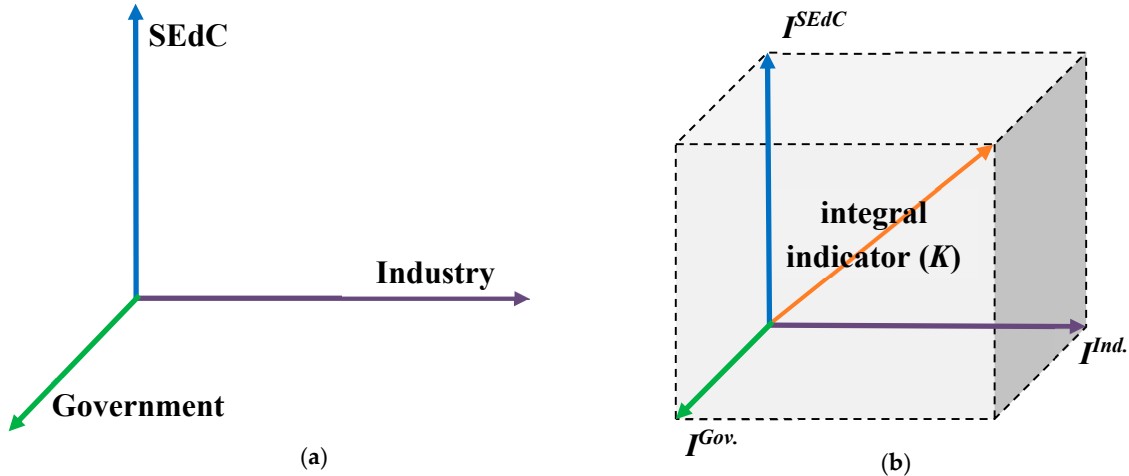

**Figure 1.** The econometric model to perform integral assessment of the economic entity innovation development (EEID) level. (**a**) axles, (**b**) the econometric model.

The proposed model allows the use of econometric modeling method, which is the most convenient modern tool for numerical calculations used for forecasting. In this model, SEdC acts as a generator of knowledge and innovative ideas, an owner of the object of intellectual property, and both government (governmental support–policy) and industry (attainment of profits–market) are interested and take active part in their commercialization. The formation of such environment requires continuous systematic work to support stable function of the innovative system of a region, based on efficient interaction between the key participants of the innovative process, with the aim of creating new business areas. The econometric model presented by N. Egorov allows the quantification of the contribution of each of the triad participants to the innovation development of economic entities of different levels, including cluster formations based on known trigonometric expressions.

Thus, economic and mathematical modeling based on the author's model allows the evaluation of the innovative activities of the Triple Helix participants in the economic development of not only a region in general, but also a separate municipality, a branch of the real economy, a territorial innovative cluster, etc. In these conditions, the EEID level is assessed on the basis of integral evaluation of the triad's (SEdC, industry, and government) contribution, through their key economic and statistical indicators in the field of innovation activity of the relevant economic entity of a region.

## 3. Formulae for Numerical Calculations

The considered approach is proposed due to the fact that the innovation potential represents not just the sum of its constituent elements, but their complex, characterized by a complex and multifaceted relationship. The advantage of the proposed integral indicator is the fact that it covers all basic constituent elements that are brought maximally to a comparable form. The selection of summary indicators was carried out, based on the following provisions:

–  the system of indicators should provide a comprehensive description of the innovation processes, including all of its main stages: "research–innovations–production–market";
–  the set of indicators should be flexible, i.e., reflect all changes occurring in the innovation sphere of the region (including resource and performance characteristics);
–  the number of indicators should be limited and associated with the peculiarities of regional statistics and its capabilities for conducting a comparable assessment of the innovation potential in the territorial context.

## 4. The Software for Numerical Calculations

As is known, several regions affect the innovative development of the Arctic. Therefore, for each region, the Triple Helix model can be applied (Figure 2). Then, to calculate the impact of a particular region on the development of the Arctic, several mathematical transformations should be performed.

Let us calculate the total resulting value of the EEID index for a particular region (according to Figure 1) using the well-known mathematical formula for determining the resulting vector of three components of a rectangular parallelepiped (the diagonal of a rectangular parallelepiped is equal to the square root of the sum of squares of its three dimensions):

$$K_j = \sqrt{\left(I_j^{gov.}\right)^2 + \left(I_j^{SEdC}\right)^2 + \left(I_j^{ind.}\right)^2} \qquad (1)$$

$I_j^{gov.}$—assessment of the impact of state projects on the innovative development of the *j*-th arctic region;

$I_j^{SEdC}$—assessment of the impact of science and education complex on the innovative development of the *j*-th Arctic region;

$I_j^{ind.}$—assessment of the impact of industries on the innovative development of the *j*-th Arctic region.

All assessments must be expressed in conventional money. On Figures 1 and 2, $K_j$ corresponds to a vector «integral indicator». We calculated the total influence of regions on the innovative development of the Arctic, using the following formula:

$$Y = \sum\nolimits_{j=1}^{m} K_j. \tag{2}$$

$Y$—total influence of regions on the innovative development of the Arctic.

To assess the impact of a particular region on the development of the Arctic, it was necessary to calculate the following indicator:

$$B = 100\% \cdot \left( \frac{K_j}{Y} \right) \tag{3}$$

The method described above was used to develop a computer software program for numerical calculations, which was certified, accordingly, by the Federal Service for Intellectual Property (Rospatent) for state registration of computer software programs [25,26].

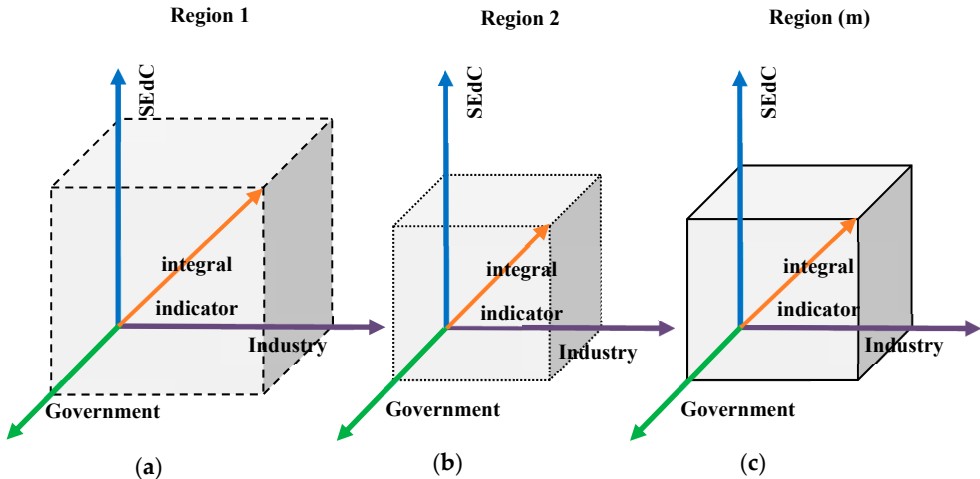

**Figure 2.** Regions affecting the innovative development of the Arctic. For three different Arctic regions (**a**), (**b**) and (**c**).

In this program, the EEID level was assessed based on the integral evaluation of the contribution of each triad participant, through their key indicators in the field of science and innovation of the corresponding region.

The program was developed with the help of the Borland Delphi 7.0 visual programming environment, using the Microsoft Access database file, and contained an executable file, a database file, and a text file of the program configuration.

The main functions of this program were as follows:

– to maintain the conducted research database and store the information in a database format in a protected mode;
– to calculate the indicators and perform the EEID level assessment;
– to generate reports to analyze and monitor the EEID for a long period of studies.

The program was implemented in a network file-server architecture and could be used for specialists of state and municipal authorities, research, and other organizations that analyze and monitor the economic development of regions. This software product could be used to develop an information and analytical system for integrated assessment and monitoring of the EEID levels, based on the Triple Helix model, incorporating the database and the analytical component. Let us note that Rospatent provided a registered database on the key indicators of research and innovation activity of the Arctic Zone of the Russian Federation (AZRF) administrative territories, for the period of 2010 to

2016, which could be used to conduct quantitative assessments of the contribution made by the SEdC, the industry, and the government, for an integral AZRF innovation development, as well as for its integral index [27].

The introduction of this information and analytical system into the regional administration system, will improve the quality of informational and methodological support in the research and analysis of innovation development, which contributes to making adequate and effective managerial decisions. Statistical indicators characterizing the research and innovation activities of the economic entities were used for the calculations. Thus, the proposed software made it possible to analyze and assess the real situations in the field of innovations that are taking place on various scales of economic entities [28].

## 5. Key Indicators of Innovation Performance

The use of direct indicators in Russia is difficult, due to the low reliability of statistics in innovation, and the lack of different indicators in a regional context. Therefore, the problem of a universal system for determining the innovation potential of regions remains relevant [29]. The choice of indicators to assess innovation in the regions is quite constrained and is associated with methodological difficulties, caused by both imperfect statistical records and methodological issues [30]. Certain difficulties are also associated with the underdevelopment of modern Russian innovation statistics, which has practically not seen much development in recent years, due to the extremely weak interest in the problem of innovation intensification, on the part of the government. In particular, state accounting of the activity of innovation infrastructure facilities has not yet been established. Consequently, till date, there are no precise data, even on their quantity, let alone any regularly collected official data, which would allow unprejudiced assessment of their performance (after all, the innovative infrastructure development is supported by federal funds as well) [31]. At the same time, the most significant disadvantage of the proposed methods is that performing a comparative assessment of the Russian Federation region, in terms of their innovative development level, shows a weak correlation of these methods, with the current problems of innovative and technological development of Russia, and the lack of clear objectives definition. That is, understanding the purposes and procedures through which the results could be used in solving practical issues of innovation management. This can be achieved if the government takes into account the technologies used in the world practice of innovation utilization, to strengthen the prerequisites for successful economic and social growth of a territory [32].

According to A.N. Lisina [33], the main problem in defining the level of regional innovation development is the lack of scientifically grounded, necessary, and sufficient number of indicators to assess the performance of regional innovative processes. The analysis of administrative requirements shows that it is necessary to identify 15–20 indicators that would form the basis of a calculation of the levels of regional innovation development, to improve the effectiveness of administrative decisions, in the field of innovation. According to E.V. Fedorova, six groups of factors are enough for quantitative characteristics of the factors that have a direct or indirect influence on the development of innovation, in the economic entities of the Russian Federation—the human capital factor; the infrastructure factor; the "Innovative Government" factor; intellectual results; the social and economic effect; and the industry specialization factor [34]. On the other hand, in a study of a specific object with a particular goal, one can get meaningful results that are useful for developing the elements of economic policy, even with the help of simple tools and limited information [35]. According to [36], the use of a significant number of indicators makes the ratings difficult to verify, as well as too cumbersome to be used as a tool for strategic management. Their simplification and alignment with target indicators of regional strategies of "smart" specialization are, thus, required [37].

Based on the above statements, one can use a simplified system of key indicators characterizing the effect of SEdC, the industry, and state contribution to the innovation development of a region, to perform an express assessment of the innovation development of this region. Under the concept of the Triple Helix model, SEdC acts as a generator of knowledge and innovative ideas, and the owner of the object of intellectual property, while both industry (attainment of profits), on the one

hand, and government (the policy of innovation support), on the other hand, are interested in their commercialization. Therefore, SEdC should produce such practical innovations that are demanded by the innovative business. According to Yu. Smirnov [38], the inventive activity makes it possible to analyze the compliance of the level of the innovation potential development, with the real sector needs of technological innovations, in two aspects—the level of inventive activity in the country and the level of practical application of the innovative activity products. In this context, the final indicator of SEdC performance is the objects of intellectual property, which are certified and registered by Rospatent, and are in demand by the innovative business represented in the form of the statistical indicator "the number of patents issued in Russia for inventions, utility models, and industrial designs, per 1000 labor potential (LP)" (indicator $I_j^{SEdC}$). Patent statistics is a unique source for analyzing the processes related to technical progress; therefore, it should become one of the possible options of a system of indicators, in the field of science and innovation, in the regions of Russia [39]. Inventive efficiency can be defined as the ability of a region to create new technologies, under the given values of the human capital and the Research and Development costs, and use it to assess the efficiency of the regional innovation system [40].

The innovation performance in industry is mainly determined by the statistical indicator "the proportion of innovative goods, works, and services in the total volume of goods shipped, and work and services performed" ($I_j^{ind.}$). The performance of innovation support by the regional executive government is determined by "the proportion of the budget spent for scientific research in the consolidated budget spending of an administrative territory of the Russian Federation" ($I_j^{gov.}$). The indicators mentioned above are published in the annual official editions by Rosstat, Rospatent, and the Federal Treasury, respectively.

As for the geometric representation of the triad relationship in the Triple Helix model (Figure 1), the combined integral index (CII) can be found as the cumulative integral contribution of the triad's key indicators that have been discussed, using the following Formula (1) [41,42], where $I_j^{SEdC}$ is the number of patents issued in Russia, for inventions, utility models, and industrial designs, un.; $I_j^{ind.}$ is the proportion of innovative goods, works, and services in the total volume of goods shipped, and work and services performed, un.; $I_j^{gov.}$ is the proportion of the budget spending for scientific research in the consolidated budget spending of an administrative territory of the Russian Federation [43].

Let us note that the names of the given indicators are based on the system of indicators of the Russian regional innovation index, developed by the National Research University "Higher School of Economics" (NRU HSE) [44], which gives $I_1$ as the key indicator of SEdC research and development performance. Accordingly, $I_2$ is the indicator of industrial innovation performance, and $I_3$ is the indicator of the government budget spending for research and innovation. The calculations are conducted on the basis of official statistics taken from Rosstat [45], Rospatent and the Federal Treasury [46,47].

## 6. Results

The regions of the Arctic Zone of the Russian Federation (AZRF) have significant innovative potential, almost not yet realized. Its efficient utilization implies the concentration of the resources to support a relatively high level of education, the development of a network of universities, academic institutions, and other federal research organizations, and the for mation of a new scientific and technical reserve. This should contribute to the creation of a system for generating knowledge, stimulating business activity, and, as a result, organizing the production of goods and services that are competitive in the global market of goods and services [48].

There are cases when it is necessary to carry out a valuation map of values, since they can be represented in different units of measures, and not just in conventional units. To do this, the following formula was used:

$$I_j^{norm} = \frac{I_j - I_j^{min}}{I_j^{max} - I_j^{min}}. \tag{4}$$

where

$I_j$—is one of $I_j^{SEdC}$ or $I_j^{ind.}$ or $I_j^{gov.}$;

$I_j^{min}$—is the minimum value equivalent to $I_j$;

$I_j^{max}$—is the maximum value equivalent to $I_j$.

In order to perform a comparative index analysis (analysis method that compares one region with another), a standard linear scaling technique (mathematical transformation of values for the convenience of visual perception of data) was used, which allows the obtainment of the normalized values of the indicators, to assess the contribution of the triad participants to an integral regional innovation development. The absolute values of the 2016 innovation indicators for the AZRF regions considered are shown in Table 1.

**Table 1.** Indicators of the innovation performance in 2016.

| AZRF Entities | $I_1$, un. | $I_2$, % | $I_3$, % |
|---|---|---|---|
| Arkhangelsk Region | 0.13 | 0.9 | 0.017 |
| Krasnoyarsk Region | 0.26 | 4.1 | 0.003 |
| Murmansk Region | 0.08 | 1.5 | 0.003 |
| The Nenets Autonomous District | 0.04 | 0.0 | 0.000 |
| The Republic of Komi | 0.07 | 0.3 | 0.069 |
| The Republic of Sakha (Yakutia) | 0.13 | 3.8 | 0.202 |
| The Chukotka Autonomous Region | 0.00 | 0.7 | 0.025 |
| The Yamalo-Nenets Autonomous District | 0.12 | 0.68 | 0.138 |
| AZRF, av. Arctic Zone of the Russian Federation (AZRF), average value | 0.11 | 1.50 | 0.06 |

Using the key performance indicators of the innovative process participants considered, one can conduct a quantitative assessment of the CII of the innovation development level in the AZRF regions, based on the Triple Helix model. The values of the AZRF CII for 2016 are shown in Figure 3.

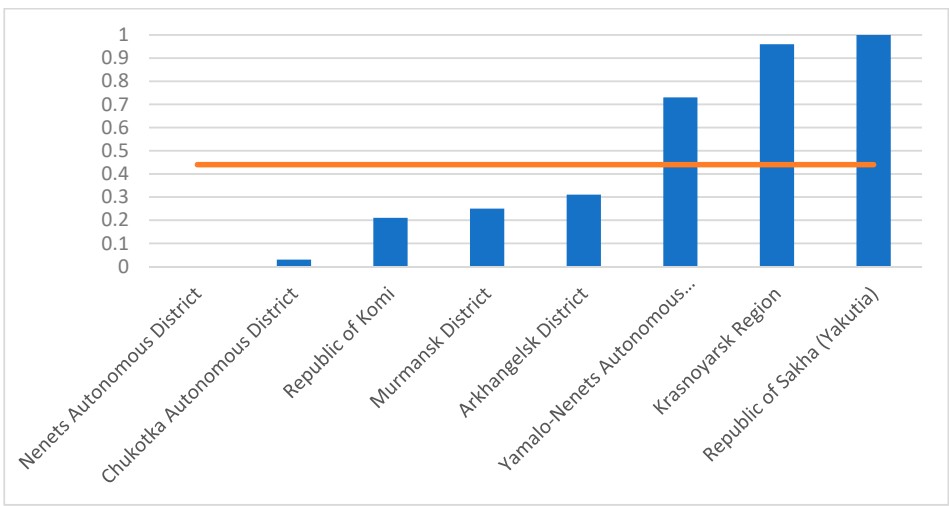

**Figure 3.** Comparative assessment of the total integral index of the innovation development level of the Arctic Zone of the Russian Federation (AZRF) regions in 2016.

In Figure 3, the Republic of Sakha (Yakutia), Krasnoyarsk Region, and the Yamalo-Nenets Autonomous District are located higher than the normalized average CII value in the AZRF (0.44). The leading places of these regions are mainly determined by the relatively similar high values of all three key indicators of the innovation performance. As is shown in Table 1, the leading place of the

Republic of Sakha (Yakutia) is determined by the good key indicators in business and governmental support of innovation, while, the second place of Krasnoyarsk Region is due to the patent activity and industry. The Yamalo-Nenets Autonomous District is characterized by a relatively high indicator in the budget spending for scientific research. In general, this is also explained by a relatively high innovative activity of organizations in the industrial sector of the Far North regions, which are primarily resource-oriented [49].

The methodology of the Triple Helix econometric model, allows the assessment of the contributions of each participant of the triad, to the integral innovation development of the AZRF, based on key indicators (Table 2).

The distribution of the contribution made by each participant of the innovative process (SEdC, industry, and regional executive government) to the integral level of regional innovation development is presented in Figure 4. According to the Figures provided, in general, the contribution of the innovation process of participants is distributed relatively evenly, throughout the AZRF, but for the Nenets and Chukotka Autonomous Districts, where a complete lack of some indicators can be observed.

**Table 2.** Contribution of the triad participants to the integral innovation development in the AZRF.

| AZRF Entities | SEdC | Industry | Government |
|---|---|---|---|
| Arkhangelsk Region | 63.4% | 26.6% | 10.0% |
| Krasnoyarsk Region | 49.6% | 49.6% | 0.8% |
| Murmansk Region | 46.1% | 51.7% | 2.2% |
| The Nenets Autonomous District | 100.0% | 0.0% | 0.0% |
| The Republic of Komi | 38.5% | 10.8% | 50.7% |
| The Republic of Sakha (Yakutia) | 21.5% | 37.7% | 40.7% |
| The Chukotka Autonomous Region | 0.0% | 58.1% | 41.9% |
| The Yamalo-Nenets Autonomous District | 36.5% | 12.4% | 51.0% |
| AZRF, average | 38.8% | 34.5% | 26.7% |

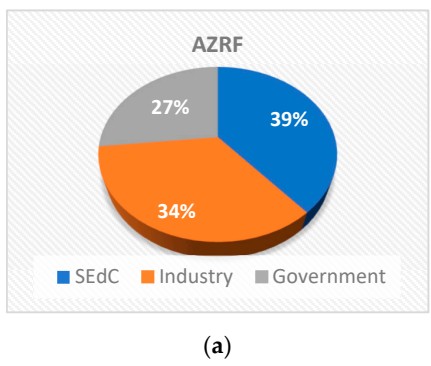

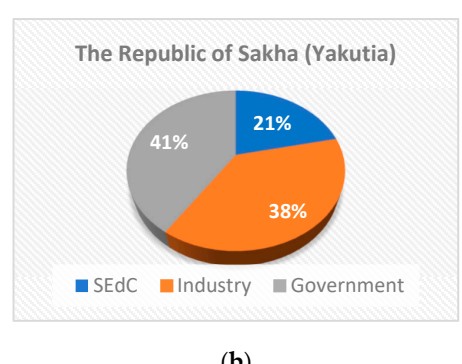

(**a**)  (**b**)

**Figure 4.** The distribution of the triad contributions to the integral innovation development in the AZRF (**a**) and its separate entities (**b**), in 2016.

## 7. Conclusions

This article describes the methodology for an integral assessment and the relevant econometric model based on the Triple helix concept, which, based on known trigonometric expressions, allow a quantitative assessment of the contributions of each of the triad participants to the integral innovation development in economic entities of different scales. The economic and mathematical modeling presented, based on the author's model, makes it possible to evaluate the innovative activity of the Triple Helix participants in the economic development within the whole region, as well as in the real sector of the economy, regional innovation clusters, etc. At the same time, the level of innovative development is measured on the basis of integral contribution assessments of the triad (science and education complex, industry, and government), using their main economic and statistical indicators in the field of innovation, in relation to a relevant regional economic entity.

The method described above was used to develop a computer software program for numerical calculations, which was certified, accordingly, by the Federal Service for Intellectual Property for state registration of computer software programs. Thus, the methodology proposed will improve the level and quality of strategic planning and management in the field of innovation development, at various entities of economic systems of different scales.

The results of a comparative quantitative assessment of the total integral index of the innovation development level at the Arctic Zone of the Russian Federation regions, have shown that, in 2016, the leading places of the Republic of Sakha (Yakutia), Krasnoyarsk Region, and the Yamalo-Nenets Autonomous District, were mainly determined by the relatively high innovative activity of the organizations in the industrial sector of the Far North regions, which were primarily resource-oriented.

In general, the results obtained using the presented methodology provided an adequate reflection of the real picture of the current state of innovation development in the Arctic regions. The comparison of the data obtained from the complex assessment of the AZRF entities' innovation development, with the other ratings, confirms the assumptions of the relative objectivity of the results obtained, using the presented author's methodology [50].

Federal Service for Intellectual Property provides a registered database on key indicators of research and innovation activity in the Arctic Zone of the Russian Federation administrative territories, for the period from 2010 to 2016, which could be used to conduct quantitative assessment of the contribution made by SEdC, industry, and government, to the integral Arctic Zone of the Russian Federation innovation development, as well as of its integral index.

The study results could be used by the state executive authorities of the AZRF regions, to adjust the existing regulatory acts in the field of innovative economy.

A promising direction for further research is related to the application of the described methodology, to address the issues of assessing the impact of innovation on the society, which implies the use of social indicators of the population, in addition to the key indicators.

Based on this methodology, the formulation of a rating of the innovative activity of the regions has been planned. Innovative activity will allow us to predict the main trends in the development of the entire Arctic region, assess the possible risks, including the impact on climate. It is fundamentally important that without including the regions of the United States, Norway, and the other developed countries in the model, it is impossible to make a forecast regarding the development of the Arctic, around the world. However, the presented methodology makes it possible to approximate the results for the whole world, with the presence of relevant statistical information.

**Author Contributions:** N.E. is responsible for general conceptualization; T.P. is responsible for investigation and interpretation of experimental results; A.Y. is responsible for formal analysis of experimental results; E.K. is responsible for conducting experimental research.

**Funding:** The research carried out with the financial support of the grant from the Program Competitiveness Enhancement of Peter the Great St.Petersburg Polytechnic University, project 5-100-2020.

**Acknowledgments:** This article was prepared with the financial support of the Ministry of education and science of the Russian Federation, within the basic part of the state task of the North-Eastern Federal University, on the project 26.8327.2017/8.9.

**Conflicts of Interest:** The authors declare no conflict of interest.

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
