# Peer review of "The Assessment of Innovation Development in the Arctic Regions of Russia Based on the Triple Helix Model"

_resources, doi:10.3390/resources8020072_

Round 1

Reviewer 1 Report

line 12 etc: "method", instead of "methodology"? see "method" in line 21

lines 18-21: can this be clarified?

line 25:  remove "method"

line 35: why "different"?

line 41: helix model [2,3], also [6,7] in line 61?

lines 50-54: clarify further U-, B-, in general, and SEC-, Ind-, or Uni- components in particular?

line 68: (SEC) again?

line 68: what's the exact model presented?

fig 1: improve it, re SEC etc

lines 100-108: what is n_i? why I=3? what is n?

lines 146-148: introduce AZRF? see line 235

line 242: briefly tell what the "index analysis" and "scaling technique" are

fig 2; improve the x label?

line 286: fig 3?

line 298: more comments on how to use the results? 

references: polish some of them; present the titles in English for e.g. line 338; remove DOI, e.g. from line 412 etc

Author Response

line 12 etc:   "method", instead of "methodology"? see   "method" in line 21

lines 18-21: can   this be clarified?

line 25:  remove "method"

line 35: why   "different"?

line 41: helix model   [2,3], also [6,7] in line 61?

lines 50-54: clarify   further U-, B-, in general, and SEC-, Ind-, or Uni- components in particular?

line 68: (SEC)   again?

line 68: what's the   exact model presented?

fig 1: improve it,   re SEC etc

lines 100-108: what   is n_i? why I=3? what is n?

lines 146-148:   introduce AZRF? see line 235

line 242: briefly   tell what the "index analysis" and "scaling technique"   are

fig 2; improve the x   label?

line 286: fig 3?

line 298: more   comments on how to use the results?

references: polish   some of them; present the titles in English for e.g. line 338; remove DOI,   e.g. from line 412 etc

Improved

Improved

Improved

Improved. Changed to   “Dissimiral”

Improved. [2,3,6,7]

Improved

Improved - SEdC

Improved

The figure 1 and   formulas (1-5) are improved

line 147 Arctic Zone   of the Russian Federation

The figure 1 and   formulas (1-5) are improved

The figure 1 and   formulas (1-5) are improved

Improved

The conclusion was   revived

The Russian articles   don’t have DOI

Reviewer 2 Report

 This paper contributed to show the methodology to evaluate the current state of innovation development in selected areas. 

 It is an interesting paper. However, as the region in the study is not familiar to me and could not get clear picture from the result. More detailed explanation about the result and relation with the features of the region could provide better understanding for the readers. 

Author Response

More detailed explanation about the result and relation with the features of the region was done to provide better understanding for the readers

Reviewer 3 Report

"Today, the development of the Arctic territories is a very hot topic. The article provides interesting results, but you should pay attention to some inaccuracies and make corrections to the description.

1. It is very difficult to find and understand what "AZRF" means. It is necessary to decipher the term.

2. Figure 1 needs some correction.

3. In formulas 1-5, each symbol should be clearly described.

4. In Figure 2 - unreadable text.

5. I would suggest authors extend the conclusion.

Author Response

1. It is very   difficult to find and understand what "AZRF" means. It is necessary   to decipher the term.

2. Figure 1 needs   some correction.

3. In formulas 1-5,   each symbol should be clearly described.

4. In Figure 2 -   unreadable text.

5. I would suggest   authors extend the conclusion.

1. Improved

2. The figure 1 and   formulas (1-5) are improved

3. The figure 1 and   formulas (1-5) are improved

4. The figure 2 is   improved

5. The conclusion was  reviewed

Round 2

Reviewer 1 Report

The presentation is improved, though minor spell check may still be required.

For example: 

Line 245: 

"I_J is one of I^SEdC or I^Ind or I^Gov" can be replaced by 

"I_J is one of I^SEdC, I^Ind and I^Gov".